# The Role of *Pseudomonas aeruginosa* in the Pathogenesis of Corneal Ulcer, Its Associated Virulence Factors, and Suggested Novel Treatment Approaches

**DOI:** 10.3390/pharmaceutics16081074

**Published:** 2024-08-16

**Authors:** Lorina Badger-Emeka, Promise Emeka, Krishnaraj Thirugnanasambantham, Abdulaziz S. Alatawi

**Affiliations:** 1Department of Biomedical Science, College of Medicine King Faisal University, Al Ahsa 31982, Saudi Arabia; 2Department of Pharmaceutical Science, College of Clinical Pharmacy, King Faisal University, Al Ahsa 31982, Saudi Arabia; pemeka@kfu.edu.sa (P.E.); 221400300@student.kfu.edu.sa (A.S.A.); 3Pondicherry Centre for Biological Science and Educational Trust, Sundararaja Nagar, Puducherry 605004, India

**Keywords:** *Pseudomonas aeruginosa*, corneal ulcer, bacterial keratitis, virulence factor, contact lenses

## Abstract

Background: *Pseudomonas aeruginosa (P. aeruginosa)*, is a diverse Gram-negative pathogen commonly associated with a wide spectrum of infections. It is indicated to be the most prevalent causative agent in the development of bacterial keratitis linked with the use of contact lens. Corneal infections attributed to *P. aeruginosa* frequently have poor clinical outcomes necessitating lengthy and costly therapies. Therefore, this review looks at the aetiology of *P. aeruginosa* bacterial keratitis as well as the bacterial drivers of its virulence and the potential therapeutics on the horizon. Method: A literature review with the articles used for the review searched for and retrieved from PubMed, Scopus, and Google Scholar (date last accessed 1 April 2024). The keywords used for the search criteria were “*Pseudomonas* and keratitis, biofilm and cornea as well as *P. aeruginosa*”. Results: *P. aeruginosa* is implicated in the pathogenesis of bacterial keratitis associated with contact lens usage. To reduce the potential seriousness of these infections, a variety of contact lens-cleaning options are available. However, continuous exposure to a range of antibiotics doses, from sub-inhibitory to inhibitory, has been shown to lead to the development of resistance to both antibiotics and disinfectant. Generally, there is a global public health concern regarding the rise of difficult-to-treat infections, particularly in the case of *P. aeruginosa* virulence in ocular infections. This study of the basic pathogenesis of a prevalent *P. aeruginosa* strain is therefore implicated in keratitis. To this effect, anti-virulence methods and phage therapy are being researched and developed in response to increasing antibiotic resistance. Conclusion: This review has shown *P. aeruginosa* to be a significant cause of bacterial keratitis, particularly among users of contact lens. It also revealed treatment options, their advantages, and their drawbacks, including prospective candidates.

## 1. Introduction

Bacterial keratitis is a serious and potentially life-threatening infection of the cornea, which is the clear front part of the eye that covers the iris and pupil. It occurs when bacteria invade and infect the cornea, leading to inflammation and damage [1]. Bacterial keratitis is primarily caused by the introduction of bacteria into the eye, often through small corneal injuries or contact lens-related issues. The most common bacterial species associated with bacterial keratitis include but are not limited to *Pseudomonas aeruginosa (P. aeruginosa)*, *Staphylococcus aureus*, *Staphylococcus epidermidis*, and *Streptococcus pneumonia* (Figure 1) [2,3]. The symptoms of bacterial keratitis include pain or discomfort, redness and swelling of the eye, excessive tearing, sensitivity to light (photophobia), and blurred or decreased vision. as well as discharge from the eye, which may be thick and yellow or green in colour [1]. Changes on the ocular surface (OS) microbiome of patients with a traumatic corneal ulcer (TCU) revealed dysbiosis in the normal microbial community on the ocular surface, which may contribute to the development and severity of corneal ulcers. At the genus level, *Pseudomonas* infection was found to be prevalent at a greater than 30% relative abundance in all individuals with TCU, while at the species level, there was a significant increase in the abundance of *Pseudomonas fluorescens* and *P. aeruginosa* in the TCU group when compared with healthy control (HC) groups, according to Kang et al. [4]. Hence, the presence of specific bacterial species and their associated functions may contribute to the inflammatory response and exacerbation of corneal ulceration in TCU [5]. Understanding the specific microbial composition and functional pathways involved in TCU can potentially lead to the development of targeted therapeutic strategies to manage and prevent these sight-threatening conditions.

There are, however, some risk factors linked with developing bacterial keratitis, and this includes wearing contact lens, as well as improper cleaning or disinfecting of these lenses or when they are worn while swimming or sleeping. Other risk factors are injury to the eye (corneal scratches, cuts, or other traumas), a weakened immune system in the immunosuppressed and immunocompromised, and any previous eye conditions which could be comorbid with pre-existing eye diseases or past surgeries. In addition to these is the long-time use of corticosteroid eye drops, which may increase susceptibility to infections [6,7,8]. Regardless of the accompanying risk factors, prompt and accurate diagnosis is crucial for the proper management and prevention of further complications. An ophthalmologist will typically perform a thorough eye examination, including slit-lamp bio microscopy, corneal cultures, and possibly other diagnostic tests [9]. Treatment will involve the use of antibiotics targeting the causative bacterial infection, and could be either in the form of eye drops or ointments. However, in severe cases, oral antibiotics or hospitalization may be deemed necessary for the management of patients with such infections. The recommendation is that patients should generally follow the prescribed treatment plan diligently as well as attending follow-up appointments to monitor the progress of healing. The consequences of delayed or inadequate treatment could lead to corneal scarring, vision loss, and, in extreme cases, corneal transplantation [10]. Generally, the risk of bacterial keratitis can be reduced by proper contact lens care, hand hygiene, and seeking immediate medical attention for any suspected eye injury or when experiencing any symptoms of an infection. This review, therefore, looks at the aetiology of *P. aeruginosa* bacterial keratitis as well as the bacterial drivers of its virulence and the potential therapeutics on the horizon.

## 2. Overview of *Pseudomonas aeruginosa* and Corneal Infections

### 2.1. Pseudomonas aeruginosa Associated Corneal Infections in Contact Lens Wearers

The formation of biofilms on the surfaces of contact lenses provides a favourable environment for the propagation and inoculation of bacterial organisms [11]. Biofilms are complex communities of bacteria embedded in a matrix of extracellular polymeric substances that offer protection and support for survival and bacterial growth. Biofilm formation could result from the extended wearing of contact lenses due to factors that include the lens material’s hydrophilicity and prolonged contact with the ocular surface. Hydrophilicity is a previously overlooked area of importance in the mechanism of corneal ulceration associated with hydrophilic (soft) contact lenses that has shed light on the role of bacterial adhesion and biofilm formation in promoting infection and ulceration. It is believed that understanding these mechanisms will be crucial in developing effective preventive measures as well as treatment strategies aimed at reducing the risk of corneal ulceration in contact lens wearers [11]. A case series reported seven patients with bilateral *P. aeruginosa* keratitis (PAK) and highlighted improper hygiene practices as potential risks associated with their wearing of contact lenses [12]. In another study that was aimed at identifying associated risk factors with poor visual acuity (VA) outcomes in *P. aeruginosa* keratitis (PAK) among contact lens (CLWs) and non-contact (non-CLWs) lens wearers suggested that the latter could be at a higher risk for more severe PAK than the former. The study indicated that the wearing of contact lenses might not be the primary risk factor for poor PAK outcomes as non-CLWs had worse VA despite not wearing lenses [13]. They did, however, since their study was a retrospective cohort study, recommend additional studies targeting larger sample sizes with prospective designs to validate their findings, thus providing comprehensive insights into the outcomes and detailed risk factors associated with PAK in both contact lens wearers and non-wearers. In addition, it has been found that the incidence rate of bacterial infections varies between users of daily disposable contact lenses, 2-week periodic replacements of soft contact lenses, and 1–3-month regular replacement contact lenses [14].

### 2.2. Uncommon Species of Pseudomonas in Corneal Infections

Other *Pseudomonas* species less commonly listed as aetiological agents of *Pseudomonas* keratitis are *Pseudomonas putida*, *Pseudomonas stutzeri*, *Pseudomonas mendocina*, *Pseudomonas ryzihabitans*, and *Pseudomonas alcaligenes.* It has been reported that corneal injury is one of the predisposing factors for *Pseudomonas* keratitis. This is sometimes reported along with other factors mentioned earlier such as age, prior ocular surgeries that could include prompt surgical interventions in complex cases, extensive ulcers, lengthier treatment times, and poor visual results [15]. A rare incidence of mixed infectious keratitis caused by *Aspergillus fumigatus*, a spore forming filamentous fungus, and *Pseudomonas koreensis* was reported in a clinical case that was described as very challenging case. This is because the managing of mixed infections requires prolonged and aggressive treatment regimens targeted at both causative pathogens [16]. Additionally, secondary scarring that could result from such mixed infections may necessitate surgical intervention to restore the patient’s vision [16]. In another report, tobramycin and the fluoroquinolone-sensitive *Pseudomonas gessardii* were reportedly isolated from patient samples of corneal ulcers. The bacterium was able to synthesise Pyoverdine, a pathogenic pigment with antioxidant characteristics that is capable of scavenging nitric oxide. It is indicated that the use of contact lenses and trauma are the two primary risk factors for *P. gessardii*-mediated corneal ulcers [17].

## 3. Modulation of Corneal Protein and Host Defence during *Pseudomonas aeruginosa* Pathogenicity

In vivo experiments involving *P. aeruginosa* keratitis in mice disclosed lysosomal enzymes from activated polymorphonuclear neutrophils (PMN), which were more essential following bacterial infection than a direct cause of injury by *P. aeruginosa* exoenzymes [18]. Also, in co-bacterial and fungal infections of human cornea, it was reported that the gene MMP9 (Matrix metalloproteinase 9) was observed to have differential expression in late stages of corneal ulcer tissue [19,20,21]. siRNA (small interfering RNA), which is able to regulate gene expression or the pharmacological blockade of HIF1A (hypoxia-inducible factor-1α), is indicated to be necessary for the successful resolution of pseudomonal infections. These infections caused less nitric oxide (NO) production, were ineffective in the killing of bacteria, and, ultimately, led to corneal perforation in a murine model of the disease. There was also a noted upregulation of HIF1A that was specific to bacterial keratitis [22]. Along with breaking down Type IV collagen in Descemet’s membrane, MMP9 is also reported to have the ability to cleave to and activate pro-IL1B, thereby causing inflammation in the cornea. However, antibiotics such as Doxycycline has been shown to have some efficacy in stopping corneal perforation associated with human microbial keratitis through the pharmacological suppression of MMP9 [21,22]. Additionally, it has been suggested that MMP13 inhibition could be utilised as a supplementary treatment in the management of microbial keratitis as well as other mucosal infections [22]. However, it is cautioned that increased MMP13 activity could cause *P. aeruginosa* keratitis through initiating the breakdown of basement membranes [23]. As a result of the aforementioned study, it has been suggested that HIF1A be considered as a potential predictive biomarker while MMPs be considered as a therapeutic target of the host to treat blinding eye diseases (Figure 2).

On the other hand, the stress hormone norepinephrine (NE) was found to be higher in the corneas of extended CL-wearing animals, while the topical administration of NE aggravated *P. aeruginosa* infection, with a resultant increase in clinical scores, neutrophil infiltration, proinflammatory cytokines, and bacterial load. This has led to the proposal of an increased corneal NE content having a role in the pathogenesis of CL-induced *P. aeruginosa* keratitis in mice [24]. It is reported that the elevation of NE in response to tissue injury appears to amplify the inflammatory response and bacterial virulence, thereby worsening corneal infection. Thus, it is suggested that the targeting of NE may provide a potential therapeutic strategy for the treatment of *P. aeruginosa* keratitis [25]. In vitro, secretory IgA (SIgA) strongly reduces the binding of *P. aeruginosa* to the damaged mouse cornea, in a dose-dependent manner [26]. An analysis of tear fluid in normal, healthy CL wearers and CL wearers with corneal ulcers showed a higher soluble IgA against lipopolysaccharide (LPS- SIgA) and that exotoxin A (ETA-SIgA) acted as a barrier to *P. aeruginosa* infectious keratitis in normal and healthy CL wearers [27].

## 4. Virulence Factors Associated with *P. aeruginosa* in Ocular Infections

*P. aeruginosa* generally has several virulence factors that contribute to its ability to establish and cause damage in its host tissue, and inclusive of those is corneal infection. Exotoxin A (ExoA), a potent exotoxin produced by the bacterium, can inhibit protein synthesis in host cells by the ADP-ribosylation of elongation factor 2 (EF-2) [28]. This disruption of protein synthesis will subsequently lead to cell death and tissue damage. ExoA, along with elastase and alkaline protease, has been previously implicated in direct and indirect contributions to *P. aeruginosa*-induced keratitis with resultant toxic effects on corneal cells, the suppression of 92 kD gelatinase (MMP9) production, and corneal protein breakdown [29]. However, in another report, ExoA was shown to have no effect on *P. aeruginosa*’s capacity to adhere to corneal wounds nor to start *Pseudomonas* keratitis, and this is stipulated to be essential for the organism to persist in the eye and ultimately cause a disease. Unlike wild-type strains, ExoA mutants were rapidly removed from the eye, leading to the subsiding of inflammation and subsequent healing of the cornea [30]. Another virulence composition of *P. aeruginosa* is pyoverdine, an iron-scavenging siderophore released by the bacterial pathogen and which is reported to control the synthesis of ExoA, which is an Endo protease, and pyoverdine itself [31]. Although the isogenic *P. aeruginosa* mutant (∆pvdE, defective in pyoverdine production) grew in vitro similarly to the original PAO1 strain and exhibited improved adherence to human cultured corneal epithelial cells (HCEC), it was, however, unable to induce keratitis in vivo [32]. *P. aeruginosa* isolates from keratitis cases exhibit significant changes in gene expression associated with bacterial secretion systems and pyoverdine metabolism compared to isolates from healthy conjunctival sacs [33]. This demonstrates that both pyoverdine and its related ExoA play crucial roles in the progression and proliferation of *P. aeruginosa* keratitis on the ocular surface (Figure 3).

### 4.1. Type III Secretion System (T3SS) of P. aeruginosa

The Type III Secretion System (T3SS) of *P. aeruginosa* is sophisticated needle-like structure that can inject virulence factors directly into host cells. T3SS contributes to the bacteria’s ability to evade host immune responses and manipulate the function of host cells, while delivering ExoU, ExoS, ExoT, and ExoY, which are effector proteins. The Type III secretion apparatus was first discovered as the protein responsible for exporting *P. aeruginosa* virulence factor into the host cell. This was done by utilizing a combination of mutagenesis, regulatory studies, expression investigations, and sequence comparisons [34,35]. T3SS effectors are typically considered to be cytotoxic, and when present in the cytoplasm of the host cell, T3SS needle proteins can cause programmed cell death. Nonetheless, when present in corneal and HeLa epithelial cells, they increase the host cells’ survival. Also, while responding to mutants lacking the T3SS, corneal cells die, but not HeLa cells, which is correlated with an increase in host factors known to regulate programmed death [36]. Many researchers are of the view that the clinical consequences of the role of T3SS in *P. aeruginosa* infections is complicated, with many clinical isolates lacking the ability to express T3SS in vitro, but still able cause disease through T3SS-independent pathways. This, however, advances the possibility of new factors affecting the bacterium’s pathogenicity as well as providing insights into the various forms of infection of *P. aeruginosa* [37]. Another contributory virulence determinant is increased levels of Cyclic di-GMP (c-di-GMP), a global bacterial secondary messenger molecule, along with the expression of the Type III secretion system (T3SS), both of which play significant roles in promoting *P. aeruginosa* corneal infections. T3SS, along with c-di-GMP, which is reported to regulate bacterial biofilm formation, contributes to the success of *P. aeruginosa* infections in the cornea [38].

### 4.2. The Role of Condensins in P. aeruginosa CL Infections

Although condensins have well-established functions in global chromosomal organization, it is unclear how they affect bacterial physiology. Structural Maintenance of Chromosomes (SMCs)-ScpAB and MksBEF are two condensins that are encoded by *P. aeruginosa* and which have been reported to be crucial to the virulence of *P. aeruginosa* during corneal infections. Increased adherence to surfaces in addition to defects in the competitive development and establishment of colonies were brought about by the inactivation of SMCs. MksB-deficient cells, on the other hand, had problems forming biofilms but no clear issues with planktonic development. Condensins have a role in the differentiation of *P. aeruginosa* and are essential for its pathogenicity, as evidenced by the fact that their inactivation significantly lowered the bacterium’s virulence [39]. Reduced virulence has been observed in SMC- and MksBEF-deficient strains, which are condensin mutants. The reduced virulence was reportedly linked to a number of cellular functions, including controlled lifestyle, primary metabolism, surface adhesion, biofilm growth, and iron and sulphur assimilation, in addition to c-di-GMP signalling and multiple virulence factors of Type 3 and 6 secretion systems. However, the mediation of the involvement of c-di-GMP signalling is partial, with no complete transcriptional responses of mutants. This, therefore, suggests the involvement of other contributory factors besides c-di-GMP that are involved in the genetic regulation of *P. aeruginosa* [40]. Reports have demonstrated the use of Diguanylate cyclase over-expressing plasmid (pYedQ) to increase the intracellular concentration of c-di-GMP, as an increase in its levels generally promotes the formation of biofilms. Resultant observations showed that, while c-di-GMP was not able to correct the protease secretion abnormalities of SMC mutants of *P. aeruginosa* in corneal infection, it was able to control several condensins responses, including both sessile behaviour and pyocyanin synthesis [40].

Countless bacteria generate pigments noted for diverse functions, one of which is defence. An example is Pyocyanin (1-hydroxy-5-methyl-phenazine), which is a blue-green pigment 90–95% of which is indicated to be exclusively produced by strains of *P. aeruginosa*. Also, the production of pyocyanin enhances the virulence of the bacterium. Pyocyanin produced by the bacterium is responsible for generating reactive oxygen species (ROS), causing oxidative stress while lowering cytoplasmic antioxidant (glutathione) concentrations within the host cells, hence leading to tissue damage.

Due to pyocyanin-induced H_2_O_2_ formation (Figure 4), a concentration-dependent loss of cellular glutathione (GSH), and an increase in oxidized GSH (GSSG), the loss of GSH takes place [41]. Also, GSH has the potential to change the way epithelial cells function and increase the cytotoxicity caused by pyocyanin through the alteration of redox-sensitive signalling events while acting as a substituted source of reducing pyocyanin equivalents [41]. Also, *P. aeruginosa* is encoded by enzymes such as polyphosphate kinase 1 and 2 (PPK1, PPK2), known to attenuate the virulence of *P. aeruginosa*. The enzyme polyphosphate kinase 1 (PPK1) is associated with virulence in ocular *P. aeruginosa* infections, while mutants that lack PPK1 are reported to have reduced ocular virulence that could be due to a reduction in the synthesis of pyocyanin [42].

### 4.3. Exoenzymes’ Pathogenesis Associated with P. aeruginosa Corneal Infection

Other virulence factors associated with *P. aeruginosa* are the various exoenzymes produced by this Gram-negative bacterium that are known to either contribute to tissue damage or play a role in evading the host immune system. An example is phospholipase C (PLC), an elastase that degrades host cell membranes and extracellular matrix components, thereby destroying host tissue. There are enzyme activities allied with *P. aeruginosa*. Amongst these are activities of cholinesterase, phosphorylcholine phosphatase, and phospholipase C, which are reported to increase when used as the only sources of either carbon, nitrogen, or both together. This is due to the high acetylcholine content in the corneal epithelium and the fact that *P. aeruginosa* causes corneal infection [43]. A virulence-promoting component of *Pseudomonas* keratitis, referred to as *P. aeruginosa* small protease (PASP), with a molecular weight of 50 kDa, was reported for the first time and identified to be the cause of corneal epithelial erosions in rabbits [44]. However, the elastase-expressing *P. putida* generated significantly higher increases in corneal erosion (SLE) scores than those of alkaline protease of *P. aeruginosa*’s elastase B and alkaline protease genes in *P. putida*, which suggests a diversity in findings while indicating that alkaline protease production could lead to only minor corneal erosion during *P. aeruginosa* keratitis, and that elastase production increases ocular disease [45]. In terms of toxins, T3SS is generally described as a complex system that can obstruct the defences of the host by injecting cytotoxins such as ExoU, ExoT, and ExoS, amongst others, into the intracellular environment [46]. Also, during an infection, *P. aeruginosa* can invade cells of the epithelia and replicate in them. This invasion of these cells will induce the formation of a ‘membrane bed’ in the invaded cells, for which three cytotoxin secretions of T3SS are reportedly needed for bedding, with Exo required for bled-niche formation. Thus, invasive *P. aeruginosa* strains induce the formation of membrane blebs which are capable of replicating and cause cell apoptosis [46]. Generally, blebs are exploited by the bacteria for intracellular replication and motility, as they are used by them for survival and movement within the cells of the host. Exoenzyme S (ExoS), an effector protein synthesised by invasive strains of *P. aeruginosa*, was discovered by Iglewski and colleagues in 1978 to be an ADP-ribosyl transferase generated by *P. aeruginosa* with distinct biochemical and enzymatic activities as compared to ExoA [47]. Heat-stable factor(s) of human corneal epithelial cell lysates were identified to contribute to T3SS induction and the secretion of ExoS, thereby contributing to the survival of and the ability to cause disease by *P. aeruginosa* [48]. On the other hand, the eukaryotic protein known as factor-activating Exoenzyme S (FAS) is needed by exoenzyme S (ExoS) in order to display ADP-ribosyl transferase activity [48]. It functions through the disruption of cellular processes by the ADP-ribosylation of specific host-cell proteins and consequently alters cell signalling and cytoskeletal dynamics. It is documented that mutant *P. aeruginosa* strains that lack T3SS or their specific components were found to be localized to intracellular perinuclear vacuoles, which are cellular compartments close to the nucleus [49]. Therefore, this might suggest that T3SS plays a role in determining the intracellular location of the bacteria, with a further suggestion that mutants that were localized to the perinuclear vacuoles gradually lose their intracellular viability over time. Mutants that lacked only the translocon structure of the T3SS were able to survive and replicate within vacuoles, thus suggesting that the translocon structure is the specific T3SS component involved in bacterial survival and replication within a distinct subset of intracellular vacuoles [50]. Reports later indicated that *P. aeruginosa* intravacuolar replication was due to ADPr activity of ExoS rather than the translocon of T3SS. The findings, however, implied that *P. aeruginosa* pathogenesis in epithelial cells could be influenced by intracellular bacteria with T3SS effectors engaged in the pathogenesis, which can be conducted without translocon-mediated trafficking across host membranes [51]. Further investigations instituted that a reduction in vascular acidity was part of the mechanism through which ExoS ADPr permitted the intracellular replication of *P. aeruginosa* [52]. Therefore, in corneal epithelial cells, the creation of membrane bleb niches and bacterial survival was dependent on the ExoS enzyme ADP-ribosyl transferase (ADP-r) rather than on the Rho-GAP activity or the membrane localization domain (MLD) [53].

## 5. Cytotoxic Strains of *P. aeruginosa*

Usually, while examining the pathological mechanism of *P. aeruginosa* infections, researchers have considered two distinct forms of *P. aeruginosa*-induced cornel illness. One of these infection forms is that of strains that may live within viable host cells, while the other is triggered by direct host-cell cytotoxicity. The cytotoxic strain PA103 was rendered avirulent because of a mutation in the ExsA gene that eliminated the strain’s capacity to generate cytotoxicity, while the virulence of the invasive strain PAO1 was unaffected by the ExsA mutation [54]. However, that a cytotoxic non-invasive *P. aeruginosa* strain was found to require ExsA for corneal epithelial colonisation and penetration showed a variation in ExsA-regulated pathogenic pathways between these two strains [55].

Exoenzyme T (ExoT) is another effector protein that interferes with host-cell signalling pathways and that could lead to cytoskeletal rearrangements, cell rounding, and detachment. The ExsA-regulated, ExoU-expressing wild-type cytotoxic *P. aeruginosa* strains do not infiltrate epithelial cells. Even though the *P. aeruginosa* ExoU mutant PA103exoU:Tn5Tc is noncytotoxic, it is non-invasive. Contrarily, a second mutation in ExsA causes an invasive phenotype, suggesting the existence of additional ExsA-regulated components involved in the inhibition of the cytotoxic *P. aeruginosa*’s ability to invade ExoT. A 53-kDa inactive version of ADP-ribosyl transferase was first described by Yahr et al. and is reported to react with antisera against exoenzyme S, with a 0.2% less enzymatic activity than ExoS despite sharing 75 % of their amino acid identity as well as both of them belonging to the same family [56,57]. Also, ExoT is the only Type III effector protein expressed by both invasive and cytotoxic strains of *P. aeruginosa* that has been so far identified and is known to prevent the invasion of cytotoxic strains into corneal epithelial cells [58]. It has been shown that cytotoxic clinical isolates of *P. aeruginosa* injured the epithelia cells of lungs in vivo while possessing ExoT but not ExoS, suggesting that the latter is not the cytotoxin responsible for the pathogenicity of *P. aeruginosa*. Finck-Barbançon et al. [59] were the first to report on Exoenzyme U (ExoU) and that it was composed of a 70-kDa protein accountable for cytotoxic phenotypes of *P. aeruginosa*. And that ExoU, ExoT, and ExoS did share identical promoter structures as well as an identical site of binding for the transcriptional activator ExsA thus demonstrating their coordinated regulation [59].

## 6. Genetic and Phenotypic Features of *P. aeruginosa*

Research on the genetic and phenotypic characterization of 46 keratitis-related *P. aeruginosa* isolates revealed that the types of ExoU or ExoS toxin gene in a strain influenced distinct protease profiles and their virulence property. Also, all serotype E (O:11) isolate strains contained the cytotoxin gene ExoU, thereby implying a strong association between this serotype and the presence of ExoU. Additionally, the rate of multidrug resistance among the isolates was significantly higher in ExoU-positive cytotoxic strains and those of serotype E (O:11) as compared to ExoS-positive invasive strains. This therefore suggests that multidrug resistance (MDR) is more commonly associated with *P. aeruginosa* corneal infections in which the causative agent possesses the ExoU toxin gene as well as belonging to the serotype E (O:11) group [60]. Other characteristics of ExoU show it to be a highly cytotoxic effector protein associated with severe infections. It possesses phospholipase activity that could lead to rapid host-cell membrane disruption, cell death, and inflammation. The active phospholipase domain is important for cytotoxic *P. aeruginosa* to traverse the multilayered corneal epithelia in vitro. Wild types or ExoU-complemented strains are commonly implicated in epithelial cell death and/or the loss of tight junction integrity as well as being involved in the correlation of transepithelial resistance (TER). However, the fact that ExoU mutants may traverse without having their TER decreased suggests the possible involvement of other mechanisms [61]. Although there is a paucity of information on the process by which ExoU is activated, it is, however, known that superoxide dismutase (Cu, Zn-containing SOD1) rather than Fe- or Mn-containing SODs, functions as a eukaryotic host co-factor that directly interacts with ExoU and is required for the induction of catalytic phospholipase activity [62].

Generally, neutrophils play a key role in immune responses and bacterial virulence factors (ExoS, ExoT, ExoU) from *P. aeruginosa* contribute to the overall inflammatory environment [63]. The aforementioned virulence factors collectively contribute to the ability of *P. aeruginosa* to adhere to and invade corneal tissue as well as its ability to evade host immune responses while causing tissue damage and resultant corneal keratitis. The severity of such infections would therefore eventually depend on the combination of virulence factors attributed to the strain of *P. aeruginosa*.

## 7. Drug-Resistance and Treatment Options of *Pseudomonas aeruginosa* in Corneal Infection

In this era of evolved antibiotic resistance, this opportunistic bacterial pathogen *P. aeruginosa* that has been associated with ocular infections has become difficult to treat, with reports spread across regions of the world, as shown in Table A1 (Appendix A). Reports as far back as the 1990s have shown an increase in resistance to topical ciprofloxacin, which is the preferred treatment for PA bacterial keratitis. However, with resistance to the antibiotic being encountered in the treatment of optical infections generally, inclusive of those of *Pseudomonas* isolates, it is cautioned that the empiric use of ciprofloxacin monotherapy be considered with care [64]. This is even more so the case in conditions that involving keratitis and corneal ulcers, as MDR could create additional problems for patients. This trend has been documented by various reports from different countries, as presented in Table A1 (Appendix A). As far as preventive measures go, it has been suggested that, in the treatment of corneal ulcers, there should be the initial antibiotic regimens should be occasionally altered [65]. It is additionally recommended that, due to resistance, antibiotics such as trimethoprim, cefazolin, and chloramphenicol should not be used as the first line of treatment, with the most effective regimen for the initial treatment of keratitis and corneal ulcers suggested to be either ceftazidime or ciprofloxacin combined with amikacin [65]. The increase in resistance to the antibiotics of choice by *P. aeruginosa* clinical isolate has been directly linked to its biofilm formation and cytotoxicity to cornea [66]. To this effect, it was reported that unilateral vision loss in a 72-year-old female corneal ulcer patient in the USA was linked to extensively drug-resistant (XDR) *P. aeruginosa* that had emanated from an artificial tears product. The whole-genome sequencing and analysis of the isolate which was resistant to the carbapenems displayed the carriage of carbapenemase genes, blaVIM-80, and blaGES-9 [67]. In another similar report, the infection of an 81-year-old patient with bacterial keratitis was attributed to an XDR *P. aeruginosa* strain that had also originated from contaminated artificial tears [68].

The XDR *P. aeruginosa* isolates were sensitive to only cefiderocol while being intermediately susceptible to colistin. Considering the cost of cefiderocol being placed at more than $2000, this drug is not affordable due to its exorbitant cost [68]. Generally, with the rise in AMR globally, alternatives to available antimicrobials (e.g., metal oxide nanoparticles, herbal extracts amongst others) are being sought after [69]. Also, clinically proven collagen-based antimicrobial agents are presently available in the market [70]. Collagen inclusion in modern clinical formulations is stipulated to function as a carrier vehicle for bioactive molecules to ensure biostability [70]. While incidences of XDR *P. aeruginosa* keratitis are particularly rare, the treatment of encountered infections is difficult. However, the treatment of severely drug-resistant *P. aeruginosa*-linked keratitis infectious with colistin as a topical eye medication together with antibiotic-soaked collagen protects proved to be successful [68]. It is therefore recommended that patients with recognised ocular and systemic risk factors be monitored closely in instituting any treatment plan, and an antimicrobial assay of the causative infection bacteria should be carried out to include all antibiotic therapeutics [71].

Globally, the susceptibility of *P. aeruginosa* corneal infection isolates to ciprofloxacin or moxifloxacin is placed at about 80%. The public health problem of the rise in antimicrobial resistance has not exempted ocular infections, especially in the US, China, and India. In a 2017–2019 retrospective observational study of microbial keratitis in a tertiary eye hospital in Harbin, northeast China, *P. aeruginosa* accounted for 3.89% of the total pathogenic isolates with a recorded 90% ciprofloxacin resistance [72]. Another study indicated that antibiotic resistance was uncommon for *P. aeruginosa* and no significant trends in resistance to fluoroquinolones were found in patients diagnosed with infectious keratitis at an ophthalmology hospital in South India between 2002 and 2013. However, over the course of 12 years, there was a noted rise in resistance to fluoroquinolones by *Staphylococcus aureus* isolates [73]. However, between 2007 and 2009, South India reported a substantial rise in the incidence of *P. aeruginosa* isolates resistant to moxifloxacin [74]. In another report from the same region, 38 patients with MDR-PA-related corneal graft infections were treated successfully with topical cefazolin 5% and ciprofloxacin 0.3% or colistin 0.19% in cases of smaller and more delicate MDR-PA graft infiltrates, while individuals with endophthalmitis needed urgent surgical interventions. Additionally, the use of lubricant ointments, compromised ocular surfaces, and the wearing of bandage contact lenses were found to be risk factors associated with MDR-PA keratitis by a tertiary eye care institution between 2007 and 2014, thereby pointing to the fact that the risk factors for the acquisition of these infections varied, with preservative-free lubricant ointments also identified as a cause or repository of such infections [75].

## 8. Drug Delivery in the Treatment of *P. aeruginosa* Cornea Infection

The T3SS inhibitor INP0341 was found to efficiently reduce the cytotoxicity of *P. aeruginosa* against human corneal epithelial cells (HCEC) without impairing bacterial growth. INP0341 did produce more reactive oxygen species and showed an increase in antimicrobial peptide expression in *P. aeruginosa*-infected HCECs, suggesting a stronger immune response [76]. Previous studies have demonstrated that topical IL-17 neutralisation improves the early prognosis for *P. aeruginosa* keratitis in C57BL/6 mice by reducing the corneal pathology, polymorphonuclear neutrophils (PMNs) inflow, and intracellular bacterial levels [77]. According to clinical scores and measurements of myeloperoxidase (MPO), the joint topical application of IL-17-neutralising Ab and ciprofloxacin in B6 mice infected with *P. aeruginosa* reduced ocular inflammation [78]. Furthermore, an anti-inflammatory effect of the mammalian b-galactoside-binding protein galectin-1 (Gal-1) has been demonstrated to include a number of mechanisms. One is the induction and differentiation of Treg, and another is the inhibition of Th1 and Th17 differentiation. Other mechanisms include the enhanced apoptosis of triggered Th1 and Th17 cells, a greater release of anti-inflammatory cytokines, and the induction of CD4+ IL-10+ T cells (Tr1). More therapeutics include those of recombinant galectin-1 (rGal-1) therapy, which has been shown to significantly improve the corneal pathology due to a decrease in neutrophil infiltration, a decrease in the production of pro-inflammatory cytokines, and a decrease in Th17 cell density in the cornea. This, therefore, shows that an endogenously generated protein like Gal-1 could be of potential therapeutic value in the fight against bacterial keratitis in vision loss and blindness in patients [79]. One more therapeutic is an antineoplastic agent, Wedelolactone (WDL). It is a key ingredient in Eclipta prostrate, which was found to target and inhibit noncanonical pyroptosis, the emergence of *P. aeruginosa* keratitis, and the release of proinflammatory cytokines [80].

Furthermore, in the past two decades, the antibiotic colistin has reemerged generally as a treatment for systemic infections caused by MDR *P. aeruginosa* [81]. In a retrospective interventional case series study from June 2011 to January 2012, treatment with 0.19% colistin as a monotherapy was found to be safe and effective in the management of MDR *P. aeruginosa* bacterial keratitis [82]. Another antibiotic tobramycin that is water-soluble has been suggested for use in rehydrating collagen shields and indicated to be an effective and convenient mode in managing *Pseudomonas* keratitis in rabbit models [83]. It has also, more recently, been shown that combinations of fortified colistin and tobramycin, administered through fortified eye drops and tobramycin-soaked collagen shields, were effective treatment options for severely resistant *P. aeruginosa* corneal ulcers in elderly patients [68]. As regards combined therapeutic options for the management of MDR infections, it was demonstrated through an in vitro *Pseudomonas* infection monoculture model of primary human corneal fibroblasts (HCFs) that fluoroquinolone ciprofloxacin, followed by levofloxacin, was highly effective in eradicating both extra- and intra-cellular bacterial keratitis as well as reducing a crucial inflammatory cytokine signal [84]. It is obvious that all *P. aeruginosa* intracellular infections must be completely eradicated by appropriate antibiotic dosage regimes to reduce the chances of re-emerging bacteria that could start fresh infection cycles [84]. However, other treatments, entailing either fluoroquinolones (ciprofloxacin or ofloxacin) or imipenem/cilastatin, were reported to have been unsuccessful in cases of multidrug-resistant *P. aeruginosa* (MDR-PA) bacterial keratitis. For these MDR-PA keratitis cases, colistimethate was recommended as an efficient alternative for their management [85].

There is the postulation that polymeric 2-methacryloyloxyethyl phosphorylcholine (MPC), a methacrylate-based polymer, has anti-biofilm capabilities. This, therefore, implies the possibility of being able to prevent bacterial attachment and growth on surfaces. To this effect, it has been reported that the contact lens material called Lehfilcon A has a special surface composition that includes MPC. Additionally, lenses with Lehfilcon A materials have been shown to be more effective at reducing *P. aeruginosa* adherence than those with other contact lens materials, thereby demonstrating that Lehfilcon could be advantageous in lowering the risk of *P. aeruginosa*-related microbial keratitis among wearers of contact lenses [86]. A silicone hydrogel contact lens incorporated with an MPC polymer layer that mimics the properties of cell membranes and ocular tissue demonstrated significant resistance to lipid deposition and cell adhesion, while potentially improving eye lubrication. These characteristics make it a promising candidate for contact lens wearers, as it may offer enhanced comfort and a reduced risk of biofouling-related problems [87].

There is also melamine, a synthetic peptide created by a combination of active sections of protamine and melittin, two antimicrobial cationic peptides. It has demonstrated bactericidal action against *P. aeruginosa* through permeabilization within five minutes and lysis within two hours post-exposure [88,89]. Derived from melamine is Mel4, which is a shorter cationic antimicrobial peptide. It has been documented that, when attached to contact lenses, it has better ocular compatibility than melamine [90]. Autolysins are released as part of Mel4’s bactericidal function, after which cells die. Comparatively, the main bactericidal function of melamine is membrane contact, which results in pore formation, membrane depolarization, and the death of cells through the release of cellular contents [91]. It was also demonstrated in animal models that contaminated contact lenses can cause microbial keratitis (MK) without a disruption in the corneal epithelium. However, contact lenses with a melamine coating exhibited a decreased occurrence of *P. aeruginosa*-related MK in vivo. Bacterial debris did adhere to the lens surface but did not result in keratitis because MK development requires living bacteria adhering to contact lenses [92]. A one-day human clinical trial to assess participants’ subjective reactions and ocular physiology during the contralateral wearing of melamine-coated and uncoated control lenses showed that melamine-coated contact lenses were worn safely by humans. Also, subjective responses and ocular physiology evaluation was carried out on rabbit models with the wearing of contralateral lenses daily for 22 days. Melamine-coated contact lenses exhibited greater rates of corneal staining, such that, even after prolonged wear, they still had strong antibacterial properties [93].

A vitamin, riboflavin (B2), is reported to serve serves as a photosensitizer in the process of corneal collagen cross-linking (CXL). It increases the production of intra- and inter-fibrillar covalent connections by photosensitized oxidation [94]. Reports of experimental findings, using a combination of this vitamin and UV light, indicated a total growth inhibition of *P. aeruginosa* while exposure to either substance alone showed no growth inhibitory effects on the bacterium [95]. Despite being a promising monotherapy or an alternative treatment for infective keratitis caused by *Fusarium solani*, CXL was found to be less successful when used alone to treat *P. aeruginosa* [96]. Although collagen-cross-linking CXL has bactericidal effects, the use of bandage contact lenses, poor perioperative hygiene, and an altered immune system could be associated risks for bacterial keratitis following treatment with CXL CVB BNHGVCX [97]. Combining CXL with antibiotic therapy for the treatment of resistant bacterial keratitis produced significant antibacterial benefits. A modified CXL (M-CXL) method, which involves concurrent CXL and frequent instillations (FI) of anti-infective agents applied once every 3–5 min for an hour, showed 100% elimination of corneal ulcers with varying aetiological factors; thus, this method is regarded as a low-risk therapy [98]. Compared to patients treated with antibiotics alone, a study of 70 patients with moderate infectious bacterial keratitis revealed that the adjunctive use of CXL along with antibiotic treatments was associated with a quicker rate to reepithelialisation, a reduced likelihood of keratoplasty, and shorter hospitalizations [99]. Despite the claimed efficacy in some clinical situations, the role of CXL in infectious keratitis is yet to be fully comprehended based on the available literature. While waiting for an emergency keratoplasty, it is suggested that CXL could be used in treating severe keratitis infections that are resistant to conventional treatment, with a suggestion that the instituting of CXL as an appropriate treatment of keratitis infections would need more large-scale randomised control trials to assess its long-term effectiveness and safety.

### 8.1. Use of Predatory Prokaryote as Antimicrobial

There are also suggestions for the possible use of predatory prokaryotes as live antibiotics to treat illnesses. To this effect, a predatory bacterium, *Micavibrio aeruginosavorus*, was reported to antagonize the growth of 7 out of 10 ocular isolates of pathogenic *P. aeruginosa*. Additionally, human corneal-limbal epithelial (HCLE) cells were not cytotoxic to this predatory bacterium and there was no increase in the secretion of the pro-inflammatory cytokines (IL-8 and TNF-α) in HCLE cells post-exposure to the predators [100]. However, a follow-up study using a rabbit ocular surface occupancy model showed that, while *M. aeruginosavorus* can affect the density of pathogens on the ocular surface and is not cytotoxic to the corneal epithelium, it does not provide an additional means of pathogen removal either over or above the host’s natural defence mechanisms [101].

### 8.2. Bacteriophages as Phage Therapy

The environment is constantly filled with viruses called bacteriophages that, without harming mammalian cells, are capable of infecting bacteria, causing lysed bacteria and subsequent bacterial death. In phage therapies using the *Pseudomonas* phage to target *P. aeruginosa* infections, it was shown to be effective in treating animal models of corneal infections. Using a strain of *P. aeruginosa* as the host, the bacteriophage KPP12 isolated from river water showed improved bacterial clearance and decreased infiltration of the neutrophil in infected cornea. As a result, it is suggested that eye-drops containing bacteriophage could be considered as a potential supplementary or alternative therapeutic option for the management of keratitis infections, particularly those that are resistant to antibiotics [102]. Another instance is that of JJ01, a lytic phage known to target *P. aeruginosa* that was identified and grouped in the family Myoviridae due to the existence of an icosahedral capsid with a contractile tail within the phage. However, there has been reports of JJ01-resistant bacterial strains. Yet, despite the emergence of these resistant strains, they exhibit noticeable physiological fitness deteriorations, thereby making them eight times more vulnerable to colistin, in addition to damaged cell membranes. It would therefore be pertinent to regard phage JJ01 as a prospective candidate for therapeutic uses in the management of *P. aeruginosa*-mediated ocular infections [103]. Screening for bacteriophages which attack clinical strains of MDR *P. aeruginosa* revealed that a bacteriophage (PPaMa1/18) which has a broad host range, strong lytic effect, and outstanding stability can be used as a potential option for the treatment of *Pseudomonas aeruginosa* infections [104].

## 9. Potential Drug Targets in *P. aeruginosa* Keratitis

The pathogenesis pf *P. aeruginosa* is facilitated by its adhesion to the site of infection and consequent biofilm formation [105]. Evidence has shown the presence of carbohydrate-binding proteins, the lectin-glycan family of lectins, and, especially, galectin in cornea epithelium for attachment [106]. Reports indicate that these proteins serve as receptors, an adhesive surface for *P. aeruginosa*, which, therefore, mediate attachment to the host during infection. Documented evidence shows that *P. aeruginosa* interacts with these specific sugars using two produced small soluble lectins, LecA and LecB, indicated to be involved with host-cell invasion and cytotoxicity, and leading to biofilm formation. According to Passos da Silva et al. [107], the addition of monosaccharides to the culture of *P. aeruginosa* inhibits its adhesive property and,, hence blocks biofilm maturation. To this end, drugs targeting the inhibition of this binding site have been found to reduce bacterial keratitis induced by *P. aeruginosa* [108]. Orally bioavailable LecB inhibitors have been found to block *P. aeruginosa* biofilm formation by preventing its host attachment. Studies have also shown that the inhibition of lectins LecA and LecB using lectin-specific monosaccharides and synthetic inhibitors reduced the bacterial formation ability of biofilms with 54 and 66% success rates, respectively [107,108].

Another drug target that could potentially promote a reduction in *P. aeruginosa* ocular infection and, hence, mitigate its biofilm formation is high mobility group box 1 (HMGB1), according to Hazlett et al. [109]. It is known that HMGB1 promotes the adverse disease effect caused by *P. aeruginosa* keratitis, leading to a poor disease outcome [110]. A previous study did show that there is an improved disease outcome after the down regulation of HMGB1 in mouse-induced *P. aeruginosa* keratitis [111]. However, HMGB1 has multiple receptors within the body; hence, inhibiting its activity might produce a complexed ripple effect because of its ubiquitous presence in many parts of the body as an immune modulator [112]. Therefore, for these targets to be clinically useful, due to the anatomy of the cornea, occasioned by prolonged treatment and the prospective of drug resistance development, drug delivery in the form of nanomedicine is advocated. This is because of the potential ocular damage that has been known to occur in *P. aeruginosa* keratosis disease if it is left untreated.

## 10. Nanomedicine in Keratitis

Topical drug delivery into the eye’s surface is the most common and accessible mode of delivery for the management of different eye illnesses, including bacterial keratitis. However, there is a limitation to the use of this method of drug delivery due to the low bioavailability of eye drops, drug resistance to antibiotics, and the restricted antimicrobial properties of universal antibiotics. Also, due to the time-consuming laboratory testing required to identify the sensitive antibiotics, the therapeutic efficacy of traditional antimicrobial drugs used in the form of eye drop-based pathogen removal regimens is compromised. Common conventional eye drops have a residence time of only 5 min and only 1–5% of the medicine penetrates the cornea due to its relative impermeability [113]. Tears on the corneal surface dilute the ophthalmic solution, and retention at the corneal surface is also crucial [114]. Nanocarriers can reduce eye tissue irritation, increase bioavailability, and improve biocompatibility with eye cells, addressing challenges with direct ocular drug administration [69]. Nanomedicine is an emerging field in the treatment of keratitis, a corneal inflammation caused by an infectious disease. Nanomedicine uses nanotechnology to improve drug delivery, increase therapeutic efficacy, and minimize negative effects. Polymer-based nanocarriers (PNCs) or polymeric nanoparticles, as well as lipid-based nanocarriers (LNCs) or liposomes, are among the most appealing types of nanocarriers. Liposomes are vesicles that can encapsulate medications, making them more stable and bioavailable. Liposomes can be made to release medications slowly, resulting in long-lasting therapeutic effects. Polymeric nanoparticles are produced from biodegradable polymers and can deliver medications in a regulated manner. They preserve medications from deterioration and increase their penetration into ocular tissue [115,116]. Zhu and colleagues [60] have reported using photodynamic bactericidal nanoparticles, specifically PαGal20-b-PGRBn, to eradicate *P. aeruginosa*. This is produced through reversible addition-fragmentation chain transfer (RAFT) polymerization, and it functions as a photosensitizer by binding to the Lec A of *P. aeruginosa* and rose bengal, specifically binding α-D-galactose species and, consequently, causing damage to DNA, RNA, proteins, and bio membranes [60]. Liposomes are frequently utilized to provide the antibacterial medications ampicillin and ofloxacin, which are used to treat infections following eye surgery. However, the liposomes made with conventional methods have a low encapsulation efficiency (EE) [117,118]. The fundamental principle of producing conventional vesicles is connected to the low encapsulation efficiency, rather than the specific type of substance selected for liposome entrapment. Supercritical assisted liposome creation (SuperLip), a one-step continuous technique based on supercritical CO_2_, was utilized to produce liposomes that were intended to carry ocular medicines like ampicillin [116].

### 10.1. Benefits of Nanomedicine in Keratitis

The advantages of polymeric and liposome nanoparticles are combined in solid lipid nanoparticles (SLNs), which provide improved stability and regulated drug release through varying modes with antimicrobials.

### 10.2. Levofloxacin

Levofloxacin-loaded SLNs with stearic acid, Tween 80, and sodium deoxycholate as co-surfactant and surfactant, respectively, promote levofloxacin entrapment and are designed utilizing a three-factor, three-level Box–Behnken design approach. Levofloxacin’s SLN had superior antibacterial activity and was determined to be non-irritating and safe for topical ophthalmic application [113]. In rabbits infected with *P. aeruginosa*, Gemifloxacin (GM) nanoparticles (NPs) that were synthesized from chitosan polymer by means of the sodium tripolyphosphate (TPP)-induced ionic gelation approach retained histological characteristics and decreased bacterial keratitis [119].

### 10.3. Ciprofloxacin

To create chitosomes, ciprofloxacin HCl-loaded reverse-phase evaporation liposomes were coated with varying molecular weights and concentrations of mucoadhesive and biocompatible chitosan polymer. An in vivo model of bacterial conjunctivitis in rabbits utilizing *P. aeruginosa* revealed that the extended-release formulation of ciprofloxacin-loaded chitosomes was responsible for controlling symptoms [120].

While nanomedicine offers significant advantages, there are challenges that need to be addressed that include the long-term safety and potential toxicity of nanoparticles. There is much-needed thorough evaluation, standardizing of manufacturing processes, and gaining of regulatory approval, all of which can be complex. Thus, further studies focused on developing more efficient, safe, and cost-effective nanomedicine solutions for keratitis, with ongoing advancements in materials science and nanotechnology driving progress, is required.

## 11. Future Research Directions

Future research would need to concentrate on developing and testing novel therapeutic options for *Pseudomonas aeruginosa* in bacterial keratitis, more so with the evolvement of multidrug-resistant (MDR) strains. Also, with the limited efficacy of conventional treatment options, novel techniques such as antivirulence tactics and phage therapy should be explored further, as they have shown to be promising potential therapies. Furthermore, extensive research is needed for the prevention of bacterial keratitis, particularly among contact lens wearers. Understanding the mechanisms of resistance and creating tailored medicines will be critical for the effective infection management and prevention of bacterial keratitis. In addition to these, future research should look at the effect of early detection and intervention in reducing the frequency and severity of bacterial keratitis.

## 12. Conclusions

This review has shown *P. aeruginosa* to be a significant cause of bacterial keratitis, particularly among users of contact lens. It also revealed treatment options as well as their advantages and drawbacks, including prospective candidates. Besides, the global health problem attributed to MDR bacterial infections is not abating and remains a concern to clinicians faced with limited treatment options. Researchers are also investigating novel approaches to treat this infection, including anti-virulence strategies and phage therapy. Regular eye check-ups are essential, especially for those who wear contact lenses or have a history of eye problems. Therefore, early detection, proper hygiene, and timely treatment remain critical in preventing and managing bacterial keratitis.

## Figures and Tables

**Figure 1 pharmaceutics-16-01074-f001:**
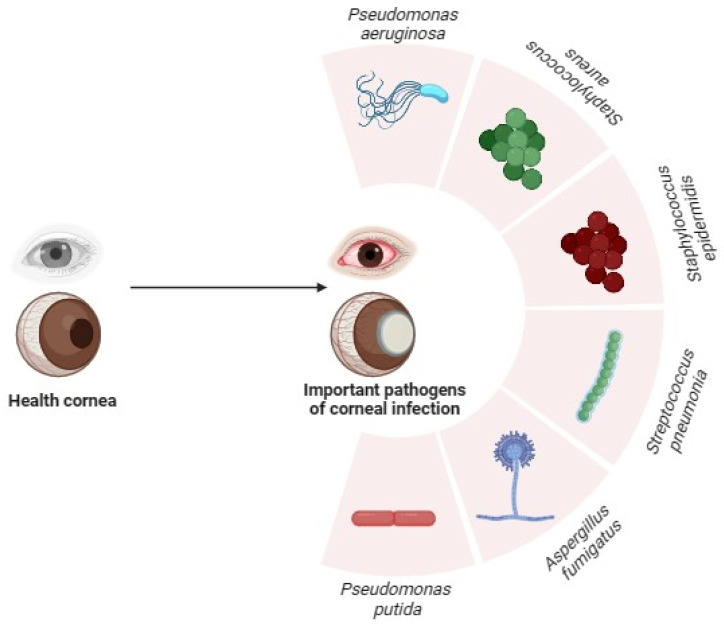
Major causative organisms responsible for development of corneal ulcers. Corneal ulcers are often caused by infections. The major causative organisms of corneal ulcers include *Pseudomonas aeruginosa*, *Pseudomonas putida*, *Staphylococcus aureus*, *Staphylococcus epidermidis*, *Streptococcus pneumoniae,* and *Aspergillus fumigatus*.

**Figure 2 pharmaceutics-16-01074-f002:**
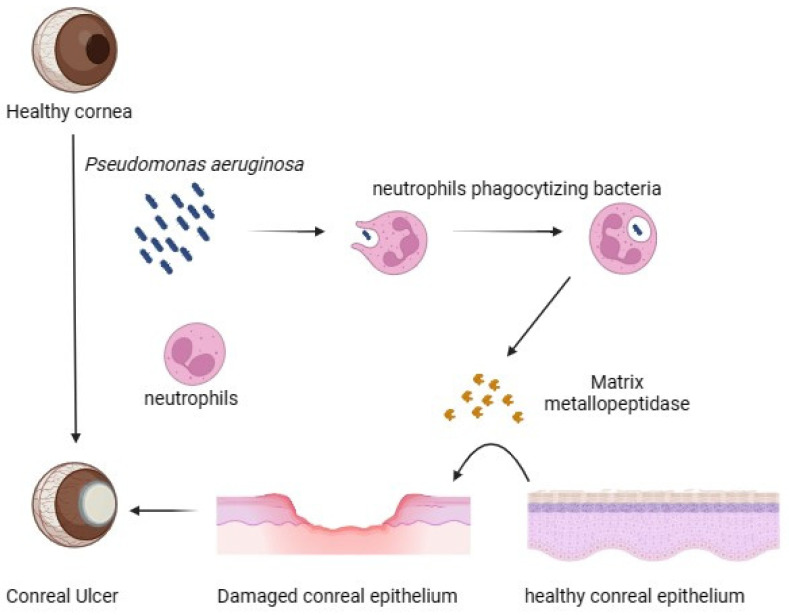
Showing modulation of host immune cell by *P. aeruginosa* and development of corneal ulcer. *P. aeruginosa* uses the pathogenic proteases and exo toxins in pathogenesis of corneal ulcer. After bacterial infection, lysosomal enzymes from activated polymorphonuclear neutrophils (PMNs) appears to be more critical in the case of *P. aeruginosa* keratitis than direct damage by *P. aeruginosa* exoenzymes. MMP9 directs the involvement of late-stage corneal ulcer tissue after *P. aeruginosa* infection of the cornea.

**Figure 3 pharmaceutics-16-01074-f003:**
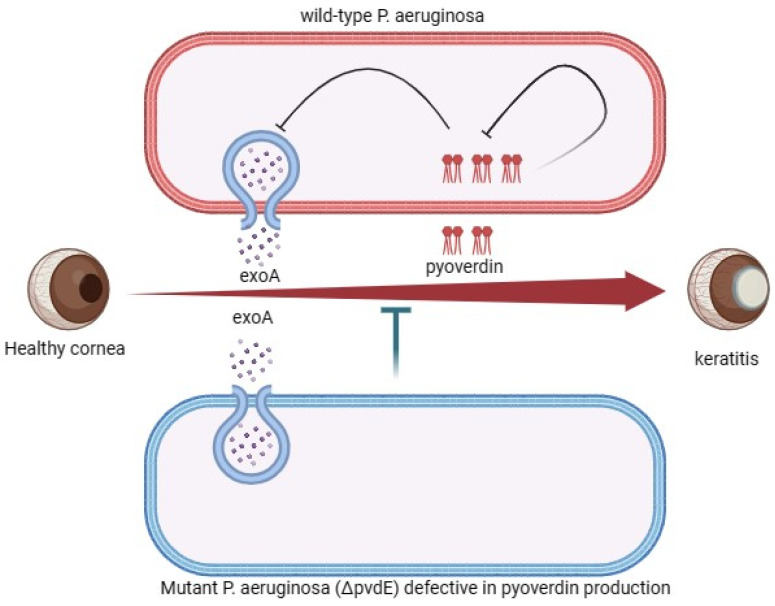
Showing the role of ExoA and pyoverdin in pathogenesis of *P. aeruginosa* in keratitis. The virulence factor ExoA’s interaction with pyoverdin enhances the progression and proliferation of keratitis on the ocular surface. *P. aeruginosa* produces Exotoxin A (ExoA), a strong exotoxin that suppresses protein synthesis in host cells by ADP-ribosylation of elongation factor 2. Pyroverdine, an iron-scavenging siderophore that regulates ExoA production and pyoverdine itself, is released by the bacterium. An isogenic *P. aeruginosa* mutant (∆pvdE, deficient in pyoverdin synthesis) showed better adhesion to human cultured corneal epithelial cells (HCEC), but it was unable to cause keratitis. Therefore, pyoverdin and its associated ExoA are essential for the development and spread of *P. aeruginosa* keratitis on the surface of the eye.

**Figure 4 pharmaceutics-16-01074-f004:**
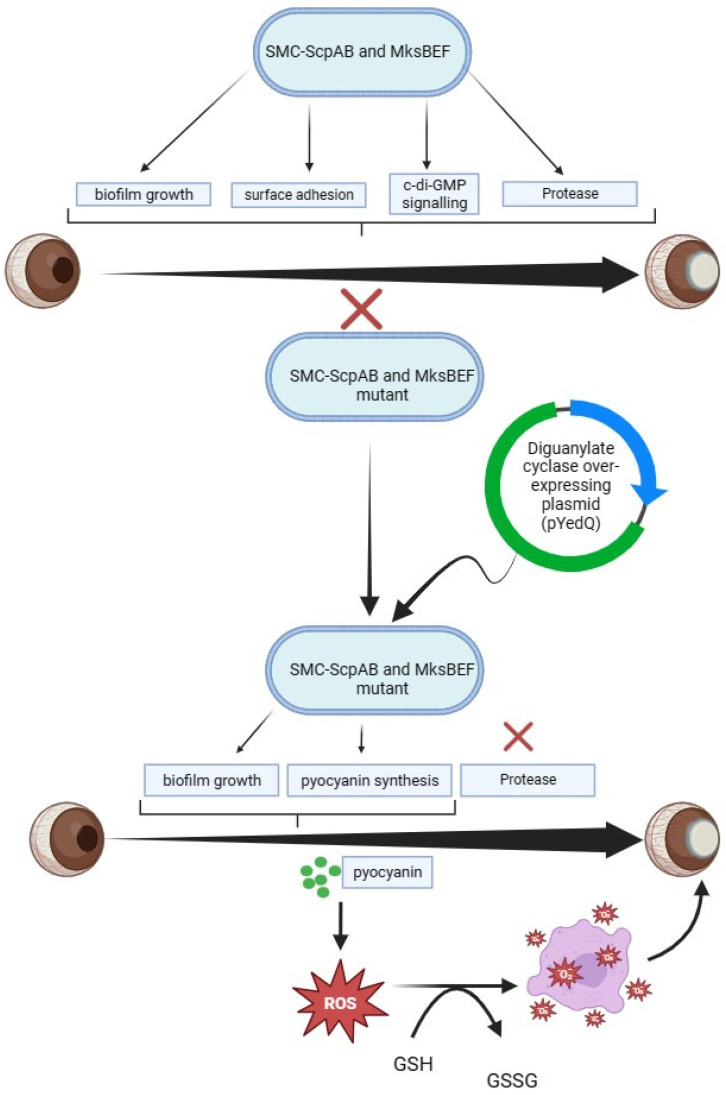
Role of c-di-GMP signalling, condensins, and pyocyanin in pathogenesis of *P. aeruginosa*-induced corneal diseases. Figure shows pyocyanin-induced H_2_O_2_ generation coupled with loss of cellular glutathione (GSH) and enhanced concentration of oxidized GSH (GSSG), with a consequent biofilm growth. 
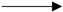
 = represents activation process, X = denotes attenuation process.

## Data Availability

Data will be made available on request.

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
