# Peer review of "The Role of Pseudomonas aeruginosa in the Pathogenesis of Corneal Ulcer, Its Associated Virulence Factors, and Suggested Novel Treatment Approaches"

_pharmaceutics, 2024, doi:10.3390/pharmaceutics16081074_

Round 1

Reviewer 1 Report

Comments and Suggestions for Authors

This manuscript is an overview of psuedomonas keratitis. It is a comprehensive review Which is valuable for publication , Of course , it has some problem to be revised. The data in this review must be categorised in order to be understand by readers in easy wayØŒthe authors should summerise the all treatment method in a table in addition each sections require a table for itself  for example  different drug treatments  must be categorised in a table therfore the reader can compare these different regimes in a  table .

I assume the authors should omit the additional information about bacterial characteristics because this part is not clinical and can be so boring for readers, in addition this part is out of the scope of this paper. The mechanism of the antibiotics are not necessary and can be omitted

Author Response

Kindly find the attached 

Reviewer 2 Report

Comments and Suggestions for Authors

The review manuscript is an appropriate summary of the situation and treatment in various Pseudomonas aeruginosa infections, including in contact lens wearers. Minor corrections are listed below. I expect a serious response from the authors.

Minor corrections

1) In the abstract and in the text of the manuscript, please append (P. aeruginosa) in brackets after the first written mention of Pseudomonas aeruginosa.

2) In the 2.1. Pseudomonas aeruginosa associated corneal infections in contact lens wearers

It has been reported that the incidence of bacterial infections also differs between daily disposable contact lenses, 2-week frequent replacement soft contact lenses and 1-3 month regular replacement contact lenses. Please add this point as well. Also, make the public aware of the causes, which include not adhering to the correct wearing time and care methods during use.

3) In the 10. Nanomedicine in keratitis

The title should be replaced by a period instead of a colon. In this section, please also add that the ophthalmic solution is diluted by tears at the corneal surface and that retention at the corneal surface is also important. Also, the heading number is 9.1 in the next sentence. The conclusion number is also incorrect.

Author Response

Dear Reviewer,

Kindly find the attached for your attention.

Reviewer 3 Report

Comments and Suggestions for Authors

This paper reviews pseudomonas aeruginosa as a cause of bacterial keratitis among contact lens users. Cleaning options are available to reduce potential infections. This study of the basic pathogenesis of prevalent P. aeruginosa strains implicates keratitis, and anti-virulence methods and phase therapy are being studied to increase antibiotic resistance. Overall, this paper is well organized, and the illustrations are clear.

Minor comments:

 Future research directions or newly emerged methods should be added to make the review comprehensive.

Author Response

(The authors gave the same response as above.)
